# Testing Some Different Implementations of Heat Convection and Radiation in the Leapfrog-Hopscotch Algorithm

Ali Habeeb Askar [1,2,3], Issa Omle [1,2,4], Endre Kovács [1,*] and János Majár [1]

1 Institute of Physics and Electrical Engineering, University of Miskolc, 3515 Miskolc, Hungary
2 Department of Fluid and Heat Engineering, University of Miskolc, 3515 Miskolc, Hungary
3 Mechanical Engineering Department, University of Technology—Iraq, Baghdad 10066, Iraq
4 Mechanical Power Engineering Department, Al-Baath University, Homs 77, Syria
* Correspondence: kendre01@gmail.com

**Abstract:** Based on many previous experiments, the most efficient explicit and stable numerical method to solve heat conduction problems is the leapfrog-hopscotch scheme. In our last paper, we made a successful attempt to solve the nonlinear heat conduction–convection–radiation equation. Now, we implement the convection and radiation terms in several ways to find the optimal implementation. The algorithm versions are tested by comparing their results to 1D numerical and analytical solutions. Then, we perform numerical tests to compare their performance when simulating heat transfer of the two-dimensional surface and cross section of a realistic wall. The latter case contains an insulator layer and a thermal bridge. The stability and convergence properties of the optimal version are analytically proved as well.

**Keywords:** heat equation; Stefan–Boltzmann law; explicit time integration; unconditionally stable numerical methods



## 1. Introduction

Diffusion of particles and Fourier-type heat conduction are omnipresent mass or energy transport processes. In the simplest linear case, they are described by the following partial differential equation (PDE):

$$\frac{\partial u(x,t)}{\partial t} = \alpha \frac{\partial^2 u(x,t)}{\partial x^2}, \tag{1}$$

where $x$, $t \in \mathbb{R}$ are the independent variables, $u = u(x,t)$ is the unknown concentration of particles or the temperature in the case of heat transfer, and $\alpha$ is the coefficient of (thermal) diffusivity. The thermal diffusivity of a material can be given as $\alpha = k/(c\rho)$, where $c = c(\vec{r},t)$, $k = k(\vec{r},t)$, and $\rho = \rho(\vec{r},t)$ are the specific heat, the heat conductivity, and the density of the material, respectively. If these coefficients depend on space, one has to use the more general equation

$$\frac{\partial u}{\partial t} = \frac{1}{c\rho} \nabla(k \nabla u), \tag{2}$$

where it is assumed that the $c$ and $\rho$ functions are positive. This equation is now valid for more than one space dimension.

According to Newton's law of cooling, a term $K(u_a - u)$ can describe (free) convective heat transfer [1], where $u_a$ is the ambient temperature (measured in Kelvin), which can be considered as independent from $u$. On the other hand, according to Stefan–Boltzmann's law [2], the heat loss of a unit surface via electromagnetic radiation can be given by a term $-\sigma u^4$. The incoming radiation, the $K u_a$ heat gain due to the nonzero temperature of the ambient air, as well as other factors such as heat generated by electric currents, can be

collected into a so-called heat source term denoted by $q$. Based on these considerations, the heat conduction Equation (1) can be extended to include heat convection, radiation, and source terms as follows:

$$\frac{\partial u}{\partial t} = \frac{1}{c\rho}\nabla(k\nabla u) + q - K \cdot u - \sigma \cdot u^4, \tag{3}$$

where the terms $q$, $K \cdot u$, and $\sigma \cdot u^4$ should also be non-negative.

Several numerical methods, which have been proposed to solve Equations (1) and (2), belong to the wide group of finite difference schemes (FDM) [3–5]. This can include different kinds of methods of lines [6], where first the space variables are discretized, and then a well-established ordinary differential equation (ODE) solver is employed to solve the obtained ODE system. These methods are usually classified as either explicit or implicit, but occasionally these two approaches are combined [7]. Implicit methods have excellent stability properties; therefore, they are commonly used for these equations [8–13]. The price for the stability is that a system of algebraic equations must be solved at each time step. This can imply very slow calculations, particularly in cases when the number of space dimensions is more than one, thus one has very large-sized and non-tridiagonal matrices. In these cases, even the most trivial explicit (Euler) time integration can be considerably faster than the implicit one [14]. Moreover, explicit algorithms can be parallelized much more straightforwardly than in the implicit methods. The main obstacle against the rise of the explicit algorithms is that they are typically only conditionally stable, i.e., the solutions can blow up if the time step size is below the so-called Courant–Friedrichs–Lewy (CFL) limit. For example, explicit Runge–Kutta methods can never be unconditionally stable [15] (p. 60). The coefficients $c$, $k$, or the diffusion constant can be highly non-uniform in space [16], e.g., when the physical properties have sharp discontinuities at the material boundaries. In these cases, the range of the eigenvalues of the system matrix has several orders of magnitude, the problem is very stiff, the CFL limit can be extremely small, thus the simulation can be unacceptably time-consuming.

There are exceptions from the above restrictions: the explicit and unconditionally stable algorithms [17–23] can be, at least partly, the solution for the above-explained problems. For example, it is reported [24] that the Dufort–Frankel scheme can be more effective in the case of fine meshes than the standard Crank–Nicolson scheme. During the last few years, we have been constructing several new explicit algorithms and have analytically proved that they are stable for the linear heat conduction equation. In those original papers [25–29], we examined the new schemes numerically as well, and found that they can outperform the widely used Runge–Kutta schemes [30] or the professionally coded MATLAB 'ode' solvers. Among these new methods, almost always the so-called leapfrog-hopscotch (LH) method has the best performance, but it can be outperformed by the original odd–even hopscotch algorithm if the system is not stiff at all. In our last paper [31], we applied some of these algorithms to Equation (3) to see whether they can be efficient under realistic conditions, e.g., when heat transfer in an insulated wall has to be calculated. However, in that paper, the convection and radiation terms were implemented in a rather ad hoc way based on our previous experiences. Moreover, no analytical proof of the stability or the convergence properties of the LH scheme has been published for the case when the equation contains anything other than the conduction term. The aim of this paper is to fill this gap. We try all possible implementations that can be simply coded. The evaluation is mostly based on numerical experiments but partly on the truncation errors and the theoretical examination of stability as well.

We are also going to use the idea contained in the nonstandard finite difference schemes initiated by Mickens [32]. The procedure of the construction of these nonstandard schemes can be found in [32] on p. 831, and in [33] on p. 1149, and one of the essential points is that the nonlinear term is advised to be treated non-locally. Therefore, we try a few different nonlocal implementations of the radiation term.

The structure of the rest of the paper is as follows. We briefly present first the leapfrog-hopscotch method, followed by the examined treatments of the convection and the radiation terms, in Section 2. In Section 3, we perform numerical tests for the convection term, and the algorithm with the best performance is analytically examined in Section 4. Further numerical experiments are presented in Section 5 in one dimension, while the algorithms are tested under realistic conditions in Section 6. The conclusions are finally summarized in Section 7.

## 2. The Examined Numerical Methods

### 2.1. The Leapfrog-Hopscotch Method for the Heat Conduction Equation

To use the leapfrog-hopscotch method, or any other odd–even hopscotch method, the space domain must be discretized using a special, so-called bipartite mesh. This means that the mesh is divided into two disjoint subsets. The nodes or cells belong to the first and second subsets that are labeled as odd and even, respectively. The main requirement is that all the immediate neighbors of the odd cells must be even and vice versa, just like on a checkerboard. We describe it in the case of a 1D interval $x \in [x_0, \ x_N]$, $L = \ x_N - x_0$ on which an equidistant grid is constructed with coordinates $x_0, x_1, \ \ldots, \ x_N$ of nodes, so $x_j = x_{j-1} + \Delta x$, $\ j = 1, \ldots, N$, $\ \Delta x = L/N$. The time domain is $t \in \left[ t^0, \ t^{\mathrm{fin}} \right]$ and it is discretized as usual: $t^j = t^0 + jh$, $\ j = 1, \ldots, T$, $\ hT = t^{\mathrm{fin}} - t^0$, where $h$ is the time step size. The mesh ratio can be defined as $r = \frac{\alpha h}{\Delta x^2}$, where $\alpha$ is defined after Equation (1). In all stages, the following version of the theta formula (obtained from the central difference formula for the space derivative) is used as a starting point:

$$u_i^{n+1} = u_i^n + r \left[ u_{i-1}^n + u_{i+1}^n - 2\theta u_i^n - 2(1-\theta)u_i^{n+1} \right], \tag{4}$$

where $\theta \in [0,1]$. The space–time structure of the algorithm is presented in Figure 1, where one can see that the neighbors $u_{i\pm1}^n$ are almost always taken into account at the time level half way between the actual old and new time levels. The first stage has the length of a halved time step, and it calculates new values for the odd nodes using $\theta = 0$, thus we have the formula

$$u_i^{1/2} = \frac{u_i^0 + r/2 \left( u_{i-1}^0 + u_{i+1}^0 \right) + h/2 \cdot q_i}{1 + r}, \tag{5}$$

symbolized by thick red arrows in the figure. Then a full time step is made with $\theta = 1/2$ for the even nodes using

$$u_i^1 = \frac{(1-r)u_i^0 + r \left( u_{i-1}^{1/2} + u_{i+1}^{1/2} \right) + hq_i}{1 + r}, \tag{6}$$

**Figure 1.** Space–time structure of the leapfrog-hopscotch (LH) scheme. Odd and even nodes are symbolized by light and dark dots, respectively.

After this, full time steps are taken alternately for the odd and even nodes with Formula (6), symbolized by blue and green arrows in the figure. Finally, a half-length time step (orange arrows) must close the calculations with $\theta = 1/2$ for the odd nodes

$$u_i^T = \frac{(1 - r/2)u_i^{T-1/2} + r/2\left(u_{i-1}^T + u_{i+1}^T\right) + h/2 \cdot q_i}{1 + r/2}. \tag{7}$$

The key point here is that the latest values of the $u$ function are always used, which means that the time indices of the node variables have to be set according to this logic, in which the figure can help. For example, when the odd node value $u_i^{1+1/2}$ is calculated, $u_{i-1}^1$, $u_{i+1}^1$, and $u_i^{1/2}$ are used, etc.

*2.2. Implementations of the Convection Term*

Until this point, the LH algorithm has been given only for the conduction and the heat source term. Now, the $-Ku$ term is also included, which is done in several ways.

1. The explicit treatment means that one calculates the increment due to the term and simply adds it to the final value of the new $u$, just as it would happen for the explicit Euler method. For example, in the case of the first stage with a halved time step, the increment is $-\frac{\Delta t}{2}Ku_i^0$, thus we have

$$u_i^{1/2} = \frac{u_i^0 + r/2\left(u_{i-1}^0 + u_{i+1}^0\right) + \Delta t/2 \cdot q_i}{1 + r} - h/2Ku_i^0. \tag{8}$$

We note that we exemplify the versions with a first-stage formula, since it is the least nontrivial due to the half-sized time step.

2. Quasi-exact treatment means that we analytically solve the ODE

$$\frac{du}{dt} = -K \cdot u, \tag{9}$$

and then take the effect of the convection terms into account in a separate calculation at the end of each stage. For example, in the case of the first stage with a halved time step, we have

$$u_i^{1/2,\ \text{temp}} = \frac{u_i^0 + r/2\left(u_{i-1}^0 + u_{i+1}^0\right) + h/2 \cdot q_i}{1 + r} \tag{10}$$

and

$$u_i^{1/2} = e^{-Kh/2}u_i^{1/2,\ \text{temp}}. \tag{11}$$

We expect that this version has an outstanding stability, since the absolute value of the solution is always smaller than the temporary value $u_i^{1/2,\ \text{temp}}$, where only the conduction and the source terms are taken into account.

3. Pseudo-implicit treatment means that the $u$ variable in the convection term is taken into account at the new time level, so Equation (4) is extended as follows:

$$u_i^{n+1} = u_i^n + r\left[u_{i-1}^n + u_{i+1}^n - 2\theta u_i^n - 2(1-\theta)u_i^{n+1}\right] - Khu_i^{n+1}. \tag{12}$$

With this, the $K$ term turns up only in the denominator, thus the first stage formula is the following:

$$u_i^{1/2} = \frac{u_i^0 + r/2\left(u_{i-1}^0 + u_{i+1}^0\right) + h/2 \cdot q_i}{1 + r + hK/2}. \tag{13}$$

Note that in our last paper [31], only this implementation was proposed.

4. Now, the $u$ variable in the convection term is taken into account at the old time level, so in Equation (12), the last term is changed to $-Khu_i^n$ This means that the $K$ term turns up only *inside* the numerator, so the first stage formula is the following:

$$u_i^{1/2} = \frac{u_i^0 + r/2\left(u_{i-1}^0 + u_{i+1}^0\right) + h/2 \cdot q_i - Khu_i^0/2}{1+r}.$$

(14)

Due to the lack of a better name, we call this version temporarily 'inside'.

5. Mixed treatment means that we make a linear combination of the last two versions (pseudo-implicit and inside) at the level of Equation (12), where the last term is changed to $-pKhu_i^{n+1} - (1-p)Khu_i^n$. The real parameter $p$ has a similar role as that which $\theta$ has for the conduction case. Now, the formulas are the following:

$$\text{First stage}: u_i^{1/2} = \frac{u_i^0 + r/2\left(u_{i-1}^0 + u_{i+1}^0\right) + h/2 \cdot q_i - (1-p)Khu_i^0/2}{1+r+pKh/2},$$

$$\text{Intermediate stages}: u_i^{n+1} = \frac{(1-r)u_i^n + r\left(u_{i-1}^{n+1/2} + u_{i+1}^{n+1/2}\right) + hq_i - (1-p)Khu_i^n}{1+r+pKh},$$

(15)

$$\text{Last stage}: u_i^T = \frac{(1-r/2)u_i^{T-1/2} + r/2\left(u_{i-1}^T + u_{i+1}^T\right) + h/2 \cdot q_i - (1-p)Khu_i^{T-1/2}/2}{1+r/2+pKh/2}.$$

We performed tests with several values of $p$, but here we present them only for three values, namely, $p = 1/3,\ 1/2,\ 2/3$.

### 2.3. Implementations of the Radiation Term

In Sections 3 and 4, it will turn out that the mixed treatment with $p = 1/2$ is the most effective; thus, when defining different treatments of the radiation term, the convection term will be taken into account that way. The radiation term will be implemented in similar ways as the convection term, but now one has more possibilities.

1. Explicit treatment:

$$u_i^{1/2} = \frac{u_i^0 + r/2\left(u_{i-1}^0 + u_{i+1}^0\right) + h/2 \cdot q_i - Khu_i^0/2}{1+r+Kh/4} - \sigma h\left(u_i^0\right)^4/2.$$

(16)

2. Quasi-exact treatment: The analytical solution of the ODE

$$\frac{du}{dt} = -\sigma \cdot u^4,$$

is

$$u(t) = \left((u_0)^{-3} + 3\sigma t\right)^{-\frac{1}{3}}.$$

(17)

This means that we have the following two sub-stages:

$$\text{Sub-stage 1}: u_i^{1/2,\ \text{temp}} = \frac{u_i^0 + r/2\left(u_{i-1}^0 + u_{i+1}^0\right) + h/2 \cdot q_i - Khu_i^0/2}{1+r+Kh/4},$$

$$\text{Sub-stage 2}: u_i^{1/2} = \left(\left(u_i^{1/2,\ \text{temp}}\right)^{-3} + 3\sigma h\right)^{-\frac{1}{3}}.$$

(18)

3. Pseudo-implicit treatment: Equation (4) is now modified as

$$u_i^{n+1} = u_i^n + r\left[u_{i-1}^n - 2u_i^n + u_{i+1}^n - 2\theta u_i^n - 2(1-\theta)u_i^{n+1}\right] - Khu_i^{n+1}/2 - Khu_i^n/2 - \sigma h u_i^{n+1}(u_i^n)^3$$

(19)

This yields

$$u_i^{1/2} = \frac{u_i^0 + r/2\left(u_{i-1}^0 + u_{i+1}^0\right) + h/2 \cdot q_i - Khu_i^0/2}{1+r+Kh/4+\sigma h\left(u_i^0\right)^3/2}$$

(20)

4. 'Inside' treatment: The last term of Equation (19) is now written as $-\sigma h (u_i^n)^4$, which yields

$$u_i^{1/2} = \frac{u_i^0 + {}^r/2(u_{i-1}^0 + u_{i+1}^0) + {}^h/2 \cdot q_i - Khu_i^0/2 - \sigma h(u_i^0)^4/2}{1 + r + Kh/4}. \tag{21}$$

5. Mixed treatment with equal share of the pseudo-implicit and inside treatments. The last term of Equation (19) is the average of the previous two cases, i.e., it is $-\sigma h(u_i^n)^4/2 - \sigma hu_i^{n+1}(u_i^n)^3/2$, which yields

$$u_i^{1/2} = \frac{u_i^0 + {}^r/2(u_{i-1}^0 + u_{i+1}^0) + {}^h/2 \cdot q_i - Khu_i^0/2 - \sigma h(u_i^0)^4/4}{1 + r + Kh/4 + \sigma h(u_i^0)^3/4} \tag{22}$$

Now, we turn our attention to the nonstandard or nonlocal treatments of the radiation term. Aiming to avoid symmetry breakings and an extensive increase in the running times, we try three different possibilities. Since the pseudo-implicit version (20) is the most successful among the treatments presented so far, we modify this version, mostly by changing one or two of the $u_i$-s in the $(u_i)^3$ product in the denominator of, e.g., Equation (20) in the following three ways.

6. Product treatment (denoted by LH PI NL prod): Instead of $(u_i^0)^3$ and $(u_i^n)^3$, we write $u_{i-1}^0 u_i^0 u_{i+1}^0$ and $u_{i-1}^n u_i^n u_{i+1}^n$, respectively.

7. Average treatment (denoted by LH PI NL av): Instead of $(u_i^0)^3$ and $(u_i^n)^3$, we write $\frac{u_{i-1}^0 + u_{i+1}^0}{2}(u_i^0)^2$ and $\frac{u_{i-1}^n + u_{i+1}^n}{2}(u_i^n)^2$, respectively.

8. In the case of the time-average treatment (denoted by LH PI NL time), there are two sub-stages. First, we calculate the effect of the diffusion and the source terms by (5)–(7) to obtain a temporary value $u_i^{\text{temp}}$. Then the time average $u_i^{\text{timeav}} = (u_i^n + u_i^{\text{temp}})/2$ is inserted into Formula (20), as follows:

$$u_i^{1/2} = \frac{u_i^0 + {}^r/2(u_{i-1}^0 + u_{i+1}^0) + {}^h/2 \cdot q_i - Khu_i^{\text{timeav}}/2}{1 + r + Kh/4 + \sigma hu_i^{\text{timeav}}(u_i^0)^2/2} \tag{23}$$

Due to the $1 - r$ factor in the numerators of, e.g., the second equation of (15), the formulas can give negative temperatures for large $r$. In these cases, large negative values of the term $(u_i^n)^3$ can arise in the denominator, which may cause instability. To avoid this, in some cases, we apply a simple trick by the following conditional statement:

$$\text{if } u_i^{n+1} < 0 \text{ then } u_i^{n+1} = 0. \tag{24}$$

When this simple trick of the prohibition of negative $u$ values is applied, it is denoted by the label 'NoNeg'. The different treatments with their notations are summarized in Table 1.

### 2.4. Methods Used for Comparison Purposes

We present three explicit methods, which are known to be unconditionally stable for the heat conduction case. However, as far as we know, they have not been applied to the case where convection and radiation are also present.

**Table 1.** The different treatments of the convection and the radiation terms and the defining equations.

| | | | Equation Number or Point |
|---|---|---|---|
| LH FullExp | Fully explicit | convection, Section 2.2 | (8) |
| | | radiation, Section 2.3 | (16) |
| LH QuasiEx | Quasi-exact | convection, Section 2.2 | (11) |
| | | radiation, Section 2.3 | (18) |
| LH PseudoImp | Pseudo-implicit, PI | convection, Section 2.2 | (13) |
| | | radiation, Section 2.3 | (20) |
| LH Inside | Inside (the numerator) | convection, Section 2.2 | (14) |
| | | radiation, Section 2.3 | (21) |
| LH Inside Noneg | Inside with the non-negative trick | convection, Section 2.2 | (14) + (24) |
| | | radiation, Section 2.3 | (21) + (24) |
| LH Mixed | Mixture of the pseudo-implicit and inside with the weight of the PI | convection, Section 2.2 | (15) |
| | | radiation, Section 2.3 | (22) |
| LH PI NL prod | Pseudo-implicit, nonlocal with product | radiation, Section 2.3, 1D | point 6 |
| LH PI NL av | Pseudo-implicit, nonlocal with space-average | radiation, Section 2.3, 1D | point 7 |
| | | radiation, Section 2.3, 2D | (39) |
| LH PI NL time | Pseudo-implicit, nonlocal with time-average | radiation, Section 2.3 | point 8 |

1. The Dufort–Frankel (DF) method [34] (p. 313) is the textbook example of explicit and unconditionally stable methods. It is a two-step but one-stage algorithm with the following formula, where the convection and the radiation terms are treated in a mixed way:

$$u_i^{n+1} = \frac{(1 - 2r)u_i^{n-1} + 2r\left(u_{i-1}^n + u_{i+1}^n\right) + 2hq_i - hKu_i^n - h\sigma\left(u_i^n\right)^4}{1 + 2r + hK + h\sigma\left(u_i^n\right)^3} \tag{25}$$

Since this algorithm is not a self-starter, $u_i^1$ must be calculated from $u_i^0$ by another method. We employ the UPFD formula [31] for this purpose:

$$u_i^{n+1} = \frac{u_i^n + r\left(u_{i-1}^n + u_{i+1}^n\right) + hq_i}{1 + 2r + hK + h\sigma\left(u_i^n\right)^3}$$

2. The alternating direction explicit (ADE) scheme is a known, but non-conventional method [23,35]. In a one-dimensional equidistant mesh, one splits the calculation into two directions, first sweeping the mesh from the left to right (using auxiliary variable *a*) and then vice versa (with variable *b*). In the case of Dirichlet boundary conditions at nodes 0 and *N*, one sets

$$a_i^n = u_i^n \,, \; i = 1, \ldots N \,, \;\; a_0^{n+1} = u_0^{n+1} \text{ and } b_i^n = u_i^n \,, \; i = N, N-1, \ldots 1 \,, \;\; b_N^{n+1} = u_N^{n+1}$$

Then, in case of pure conduction, the following equations are solved from left to right and from right to left, respectively:

$$\frac{a_i^{n+1} - a_i^n}{h} = \frac{\alpha}{\Delta x^2}\left(a_{i-1}^{n+1} - a_i^{n+1} - a_i^n + a_{i+1}^n\right)$$

and

$$\frac{b_i^{n+1} - b_i^n}{h} = \frac{\alpha}{\Delta x^2}\left(b_{i-1}^n - b_i^{n+1} - b_i^n + b_{i+1}^{n+1}\right)$$

Since, on the right hand side of these formulas, both $a_i$ and $b_i$ are taken into account 50–50% in the old and new time level, it is plausible to use the mixed treatment of the convection and radiation term here, too. With this, the explicit expressions are the following:

$$a_i^{n+1} = \frac{(1 - r - hK/2)a_i^n + r\left(a_{i-1}^{n+1} + a_{i+1}^n\right) + hq_i - h\sigma\left(a_i^n\right)^4/2}{1 + r + hK + h\sigma\left(a_i^n\right)^3/2}$$

$$b_i^{n+1} = \frac{(1 - r - hK/2)b_i^n + r\left(b_{i-1}^n + b_{i+1}^{n+1}\right) + hq_i - h\sigma\left(b_i^n\right)^4/2}{1 + r + hK/2 + h\sigma\left(b_i^n\right)^3/2} \tag{26}$$

The final values are the simple averages of the two half-sided terms: $u_i^{n+1} = \left(a_i^{n+1} + b_i^{n+1}\right)/2$. We note that for non-uniform meshes, the ADE method loses its fully explicit character, and matrix calculations would be necessary, so in Section 6 it is not used.

3. The original odd–even hopscotch (OOEH) algorithm has been known for half a century [36]. Its time–space structure is presented, e.g., in [25,37]. It uses the usual FTCS formula (based on explicit Euler time discretization) at the first stage and the backward time central space (BTCS) formula (implicit Euler time discretization) in the second stage. We now adapt it to our case in a way where the convection term is always taken into account at the new time level, while the radiation term is treated first explicitly and then in the pseudo-implicit way. The used formulas are the following:

First stage:

$$u_i^{n+1} = \frac{(1 - 2r)u_i^n + r\left(u_{i-1}^n + u_{i+1}^n\right) + hq_i - h\sigma\left(u_i^n\right)^4}{1 + hK}$$

Second stage:

$$u_i^{n+1} = \frac{u_i^n + r\left(u_{i-1}^{n+1} + u_{i+1}^{n+1}\right) + hq_i}{1 + 2r + hK + h\sigma\left(u_i^n\right)^3} u_i^{n+1} = \frac{u_i^n + r\left(u_{i-1}^{n+1} + u_{i+1}^{n+1}\right) + hq_i}{1 + 2r + hK + h\sigma\left(u_i^n\right)^3}$$

## 3. Numerical Experiments for the Convection Term

In Sections 3 and 5, our aim is to mathematically examine the algorithms, thus the units of the quantities are omitted. In this paper, the MATLAB software was used for all numerical calculations. The accuracy is characterized by the usual $L_\infty$ error, which compares the accurate reference value $u_i^{\text{ref}}$ and the result $u_i^{\text{num}}$ obtained by the actual numerical method at the final time $t^{\text{fin}}$:

$$Error = \max_{1 \le i \le N} \left| u_i^{\text{ref}}(t^{\text{fin}}) - u_i^{\text{num}}(t^{\text{fin}}) \right|. \tag{27}$$

This error is calculated as a function of the time step size $h$. More concretely, the error is first calculated for a very large $h$, then this is repeated with time step sizes subsequently decreased by a factor of 2 until small error values are reached.

Experiment 1—moderately strong convection. We start with a case where the convection coefficient is not very large, $K = 3$, while $\alpha = 1$. The initial and the heat source functions are the following:

$$u^0(x) = 5e^{x/4}[2 + \sin(2x)] \ , \ q(x) = 10[1 + \sin(5x + 0.2)]. \tag{28}$$

The considered domain is $x_0 = 0$, $x_N = 5$, $t^0 = 1$, $t^{\text{fin}} = 2.2$. The space step size is $\Delta x = 0.0125$. The fixed Dirichlet boundary conditions are simply the initial values at the boundaries $u^0(x_0)$ and $u^0(x_N)$. We employed the ode15s solver to gain the reference solution. It is a professionally coded, variable-step, variable-order solver based on the

(implicit) numerical differentiation formulas of orders 1 to 5. It is used with a very narrow tolerance ($10^{-10}$) to create an accurate reference solution, which is utilized in Equation (27) to compute the maximum error. These errors are presented as a log–log graph in Figure 2. The thin dashed line is exactly proportional to the second power of the time step size, and one can clearly see that the, e.g., ADE and the symmetric mixed version of the LH method have second-order convergence.

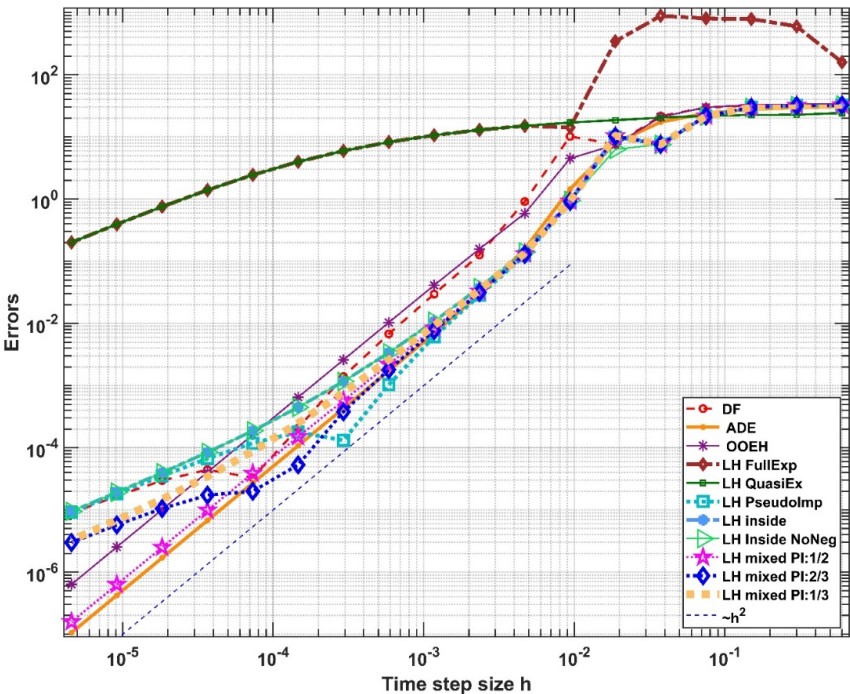

**Figure 2.** The $L_\infty$ errors as a function of the time step size $h$ for moderate $K$ (Experiment 1).

In Figure 3, we also present the initial and final temperatures for the LH method for $h = 9.38 \cdot 10^{-3}$. In this case, the maximum error of the "inside" implementation is 0.949, while it is 0.926 for the mixed method. One can observe that the numerically obtained functions are very similar to the reference solution, even in the case of these rather large (absolute) errors.

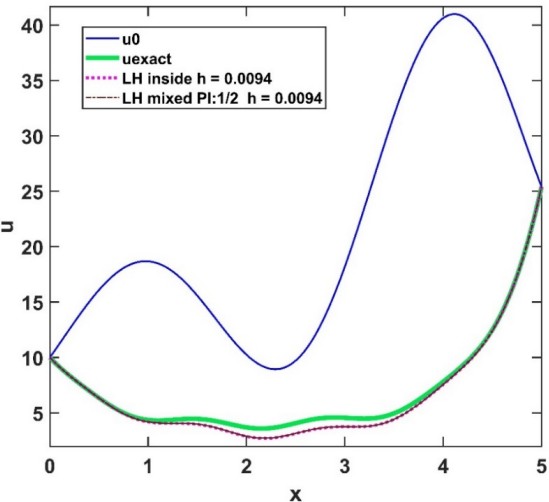

**Figure 3.** The temperature $u$ vs. the $x$ variable in the case of Experiment 1 for the initial function $u^0$, the reference solution at $t^{\text{fin}}$, and the LH method for $h = 2 \cdot 10^{-3}$ when the convection term is inside the numerator, and when it is implemented in a mixed way.

Experiment 2—very strong convection. Now, a case is considered where the convection coefficient is much larger, $K = 100$, while $\alpha$ is still one. The initial and the heat source functions are the following:

$$u^0(x) = e^x[1 + \sin(2x)] \ , \ q(x) = 1000[1 + \sin(8x + 0.2)].$$

The considered domain and all other circumstances are the same as in the previous experiment, but the space step size is $\Delta x = 0.005$. The error curves are presented in Figure 4.

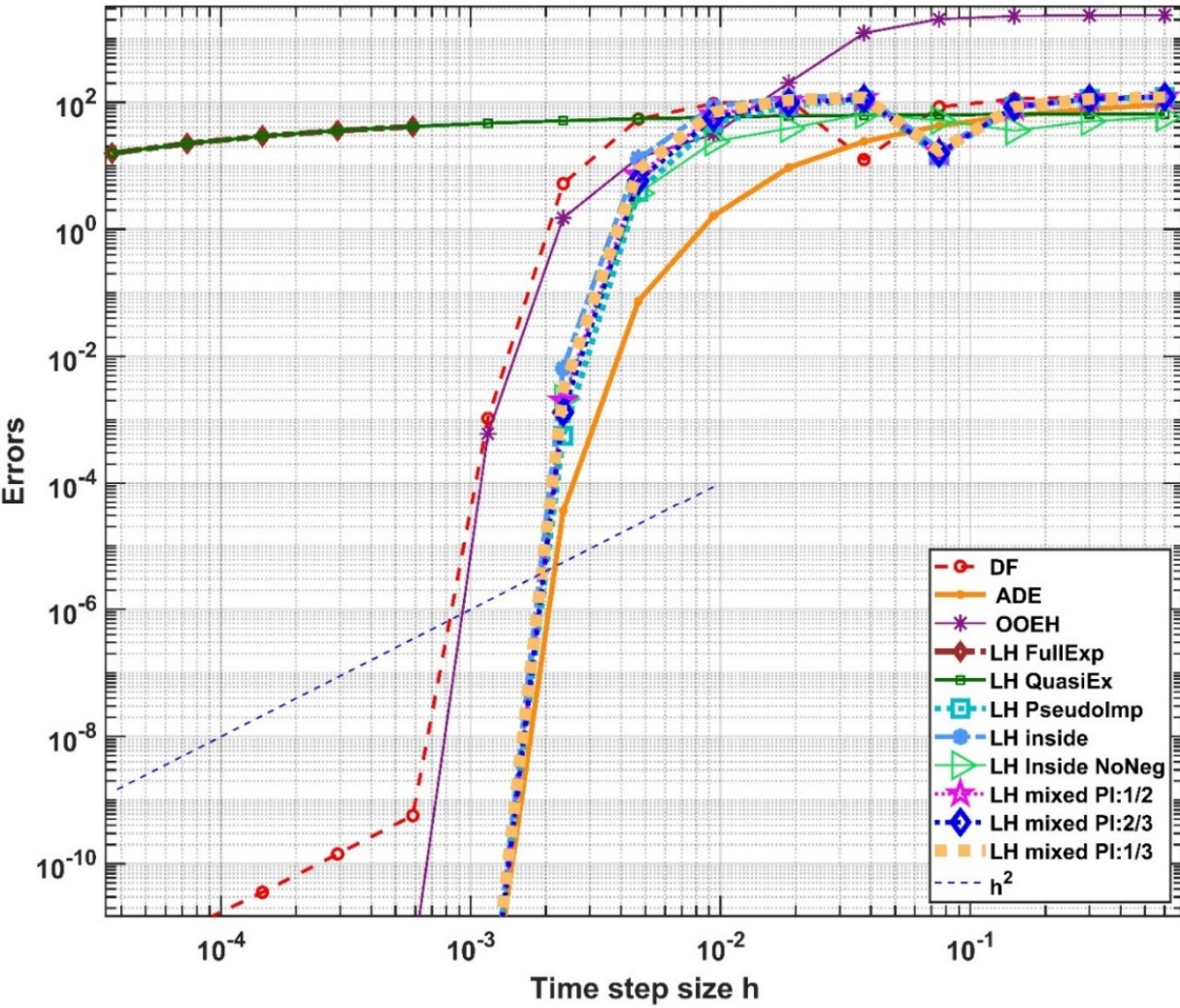

**Figure 4.** The $L_\infty$ errors as a function of the time step size $h$ for large $K$ (Experiment 2).

In Figure 5, we also present the initial and final temperatures for the LH method for $h = 2 \cdot 10^{-3}$, when the maximum error of the "inside" implementation is 3.31, while it is 1.35 for the mixed method. The numerical solutions are very similar to the reference solution, even in the case of these rather large errors.

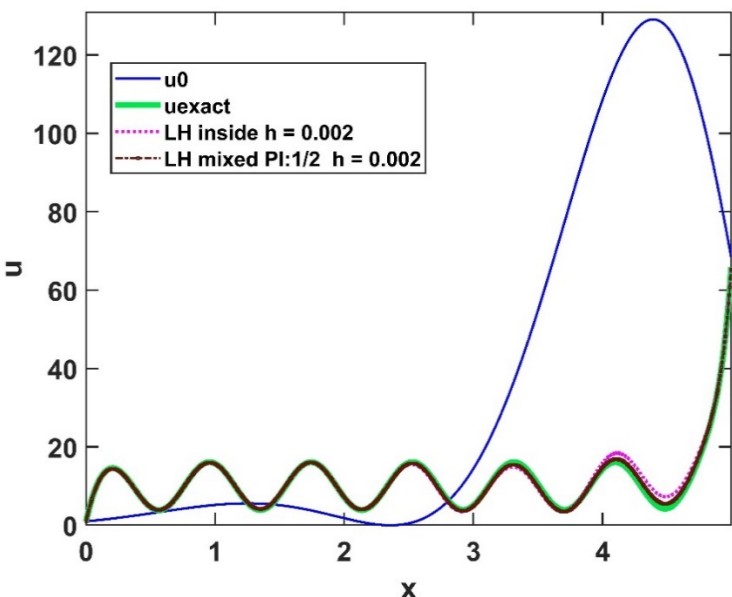

**Figure 5.** The temperature $u$ as a function of the $x$ variable in the case of the initial function $u^0$, the reference solution at $t^{\text{fin}}$, and the LH method for $h = 2 \cdot 10^{-3}$ when the convection term is inside the numerator, and when it is implemented in a mixed way.

## 4. Analytical Results for the Conduction–Convection Case

### 4.1. Consistency

We calculate the truncation error in the most common way, where subscripts with $t$ or $x$ are for differentiation with respect to the time or space variables, respectively. The exact solution is substituted into the equation containing the finite difference formulas, and

$$\tau = D_t^+ u - D_x^2 u + Ku \tag{29}$$

is the truncation error. In our case,

$$D_t^+ u = \frac{u(x_i, \ t^n + \Delta t) - u(x_i, \ t^n)}{\Delta t}$$

is the usual first order forward difference operator for the time derivative. However, the second order central difference operator for the space derivative

$$D_x^2 u = \alpha \frac{u(x_i - \Delta x, \ t^n + \Delta t/2) - u(x_i, \ t^n) - u(x_i, \ t^n + \Delta t) + u(x_i + \Delta x, \ t^n + \Delta t/2)}{\Delta x^2}$$

and the convection term

$$Ku = Kpu(x_i, \ t^n + \Delta t) + K(1 - p)u(x_i, \ t^n)$$

contain different time levels. The Taylor series expansion of $u$ is substituted into Equation (29). The function $u$ is supposed to be the exact solution, thus the identity $u_t = \alpha u_{(2x)} - Ku$ is used. The obtained truncation error is

$$\tau = \tau_0 - 1/2K(2p - 1)(Ku - \alpha u_{xx})h + \frac{1}{8}\left(K + \frac{2\alpha}{\Delta x^2}\right)u_{tt}h^2 + \frac{1}{24}u_{(3t)}h^2, \tag{30}$$

where $\tau_0$ is the discretization error of the *standard* central difference formula (used in, e.g., the FTCS method):

$$\tau_0 = -\frac{\alpha}{12}u_{(4x)}\Delta x^2 - \frac{\alpha}{360}u_{(6x)}\Delta x^4 - \frac{\alpha}{20160}u_{(8x)}\Delta x^6 + \text{h. o. t.} \tag{31}$$

The highest order term cancels out only if $2p = 1$. In the case of traditional methods, such as the FTCS scheme, the discretization error of the $D_t^+$ and $D_x^2$ operators depend on only $\Delta t$ and $\Delta x$, respectively. In those cases, the space- and time-dependent terms in the truncation error can be clearly separated. This does not hold in our case, where there is a term containing the ratio of $\Delta t$ and $\Delta x$. We can summarize these results in the following theorem.

**Theorem 1.** *When applied to PDE $u_t = \alpha u_{xx} - Ku$ , the algorithm given by Formulas (15) is conditionally consistent. The order of temporal consistency/convergence is two if and only if $p = 1/2$. In this case, if the space and time step size tend to zero $\Delta x \to 0$, $h \to 0$, such that $\frac{h}{\Delta x} \to 0$, then the error of the numerical solution compared to the analytical solution of the PDE tends to zero with $h^2$.*

*4.2. Stability*

To perform a von Neumann stability analysis, the $u$ values in the expression of the general (i.e., not the first and the last) step of the algorithm

$$u_i^{n+1} = u_i^n + r\left[u_{i-1}^{n+1/2} + u_{i+1}^{n+1/2} - u_i^n - u_i^{n+1}\right] - Kh\left(u_i^n + u_i^{n+1}\right)/2 \tag{32}$$

must be replaced by the appropriate errors $\varepsilon$. Then, the error functions must be decomposed into Fourier series, as follows:

$$\varepsilon_i^n = \sum_m E_m(t)e^{Ik_m x}, \quad \varepsilon_{i\pm 1}^{n+1/2} = \sum_m E_m(t + h/2)e^{Ik_m(x\pm\Delta x)}$$

and,

$$\varepsilon_i^{n+1} = E_m(t + h)e^{Ik_m x}$$

where $E_m(t)$ is the amplitude of the $m$-th term $e^{Ik_m x}$ in the Fourier series of the error, and $I$ is the imaginary unit $\sqrt{-1}$. For brevity, the $m$ index is omitted, and the notations $\gamma = k_m\Delta x$ and $\kappa = Kh/2$ are introduced. Now, the following relations can be written

$$\varepsilon_{i-1}^{n+1/2} + \varepsilon_{i+1}^{n+1/2} = E(t + h/2)e^{Ikx}\left(e^{-I\gamma} + e^{I\gamma}\right) = 2E(t + h/2)e^{Ikx}\cos\gamma.$$

For the sake of simplicity, we denote $E(t)$, $E(t + h/2)$ and $E(t + h)$ by $g^0 = 1$ , $g^1 = g$, and $g^2$. Performing all of these substitutions and simplifying with $e^{Ik_m x}$, we obtain a second-order algebraic equation for the factor $g$:

$$g^2 = 1 + r\left[2g\cos\gamma - 1 - g^2\right] - \kappa\left(1 - g^2\right) \tag{33}$$

If the solutions

$$g_{1,2} = \frac{r\cos\gamma \pm \sqrt{r^2\cos^2\gamma + 1 - r^2 - \kappa(2r + \kappa)}}{1 + r + \kappa} \tag{34}$$

of this equation are in the closed interval $[-1, 1]$ for arbitrary parameter values $r$, $\kappa$, and $\gamma$, then the errors cannot be amplified, which means unconditional stability. We performed a thorough investigation of these two functions and obtained that it is indeed the case, so the following theorem can be stated.

**Theorem 2.** *If $p = 1/2$, then Algorithm (15) is unconditionally stable for the equation $u_t = \alpha u_{xx} - Ku$ for arbitrary values of K and $\alpha = k/(c\rho)$.*

## 5. Numerical Experiments for the Radiation Term in 1D

*5.1. Verification with an Analytical Reference Solution*

Experiment 3. The following function

$$u^{\text{exact}}(x, t) = te^{x-t} \tag{35}$$

is a simple exponential analytical solution [28] of the one-dimensional conduction–convection–radiation equation if $\alpha = 1$, $K = 2$, and if the heat source function is $q(x, t) = \sigma t^4 e^{4x-4t} + e^{x-t}$. The initial $u^0$ function and the time-dependent Dirichlet boundary conditions are calculated straightforwardly by substituting the initial time and boundary space values to the analytical solution (35), respectively. The computational domain is $(t, x) \in [1, 3] \times [0, 3]$, while the space step size is $\Delta x = 0.005$. We use an extremely large, non-physical value $\sigma = 10$ for the radiation coefficients in order to obtain information about the behavior of the schemes in general conditions. With this, the largest value of the term $\sigma \cdot u^4$ is above $1.2 \cdot 10^4$, which means that the nonlinearity is very strong. The errors as a function of the time step size are presented in Figure 6. As one can see on the left side of the figure, the errors tend to very small numbers for decreasing time step sizes, so the algorithms are verified. However, some of the algorithms are unstable, and for medium time step sizes (around $h = 10^{-2}$), their markers are missing from the figure. These are the DF, ADE, the LH fully explicit, LH inside, and LH mixed schemes. The OOEH is not literally unstable but it produced errors which are as large as $10^3$, thus it is also not robust in this strongly nonlinear case. The error of the other methods is limited, even for these large step sizes in this experiment, which means they have very good stability properties. The nonlocal treatment of the radiation term with the product of the neighbors performs the best, but the other nonlocal and the pseudo-implicit treatment is also successful. In the case of the 'inside' and 'mixed' versions, the non-negative condition (24) helped to maintain the stability, but they are still outperformed by the nonlocal and the pseudo-implicit versions.

*5.2. Results in Case of a Numerical Reference Solution*

Experiment 4. The coefficients are set to $\alpha = 1$, $K = 10$, and $\sigma = 10^{-7}$. The initial and the heat source functions are the following:

$$u^0(x) = 100 + 5e^{x/2}\sin(x/2)[5 + \sin(4x)] \ , \ q(x) = 10{,}000\sin^{14}(6x + 0.2). \tag{36}$$

The considered domain is $x_0 = 0$, $x_N = 6$, $t^0 = 1$, $t^{\text{fin}} = 1.2$. The space step size is $\Delta x = 0.01$. The fixed Dirichlet boundary conditions are $u^0(x_0)$ and $u^0(x_N)$, and the reference solution is served by ode15s, as in Section 3. The errors are presented in Figure 7. The three algorithms with the non-negative trick (24) produced the same errors as their counterparts without the trick, so we omitted them from the figure. The temperature variable is plotted as a function of space in Figure 8 for the initial and the reference solution, as well as for the LH method with the nonlocal product treatment for $h = 2 \cdot 10^{-3}$, when its maximum error is 5.72.

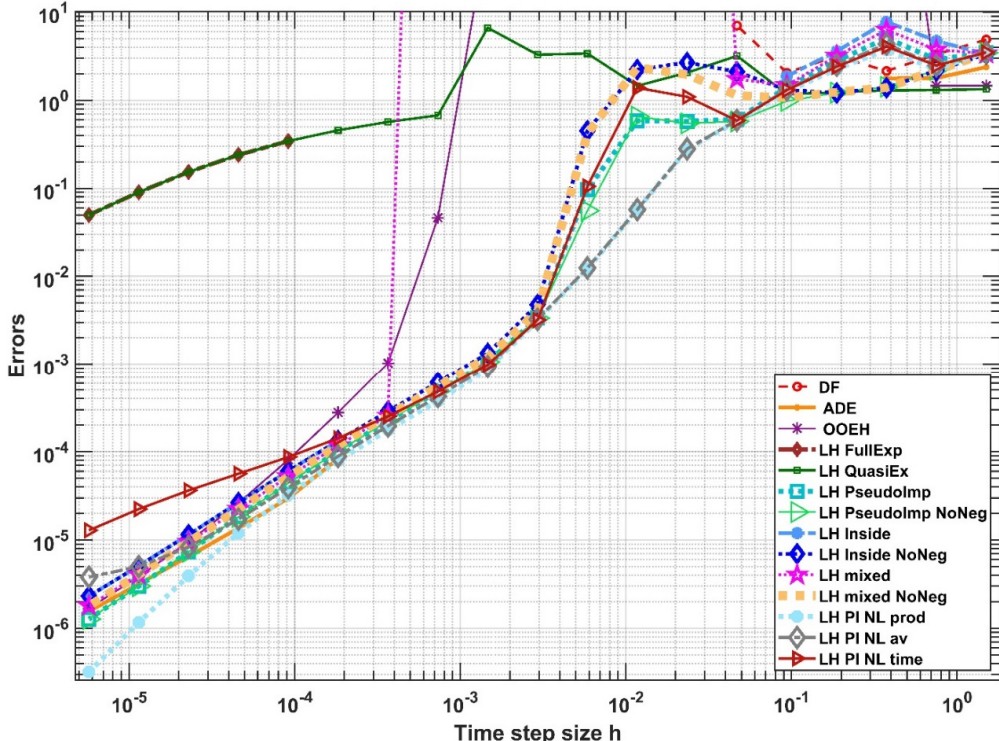

**Figure 6.** The $L_\infty$ errors vs. the time step size for the numerical algorithms defined in Sections 2.3 and 2.4. We emphasize that the different treatments now refer to the radiation term.

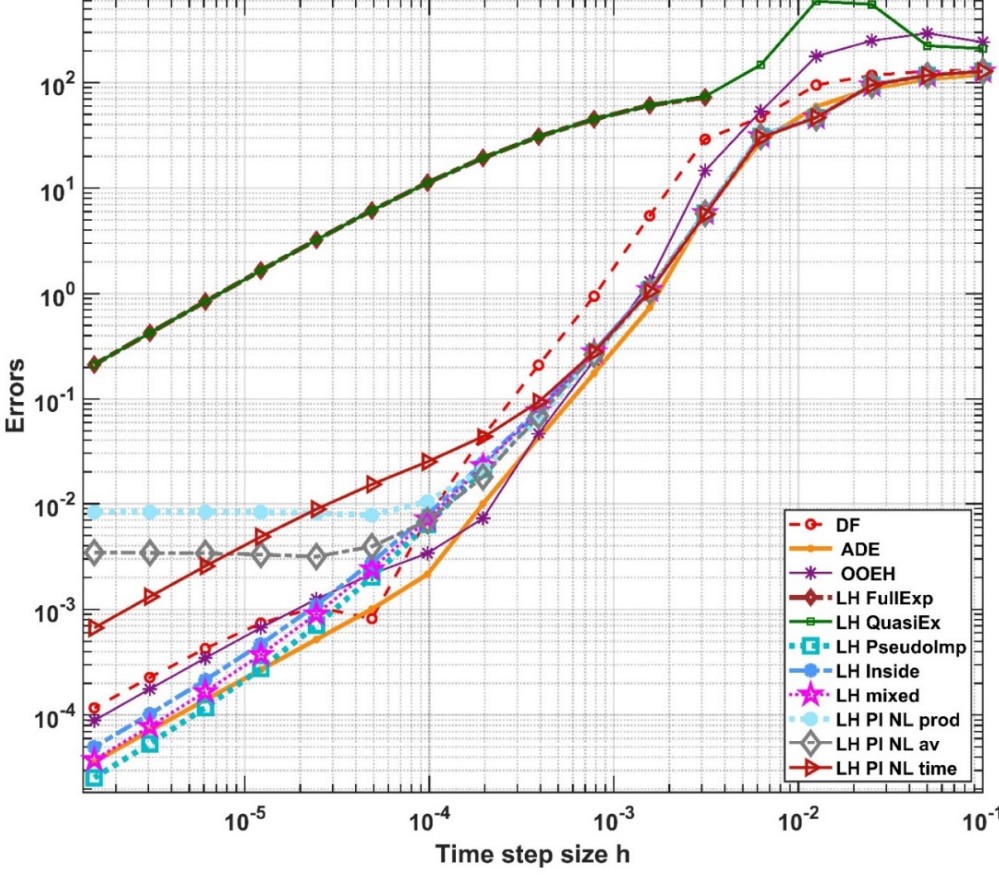

**Figure 7.** The $L_\infty$ errors as a function of the time step size $h$ (Experiment 4).

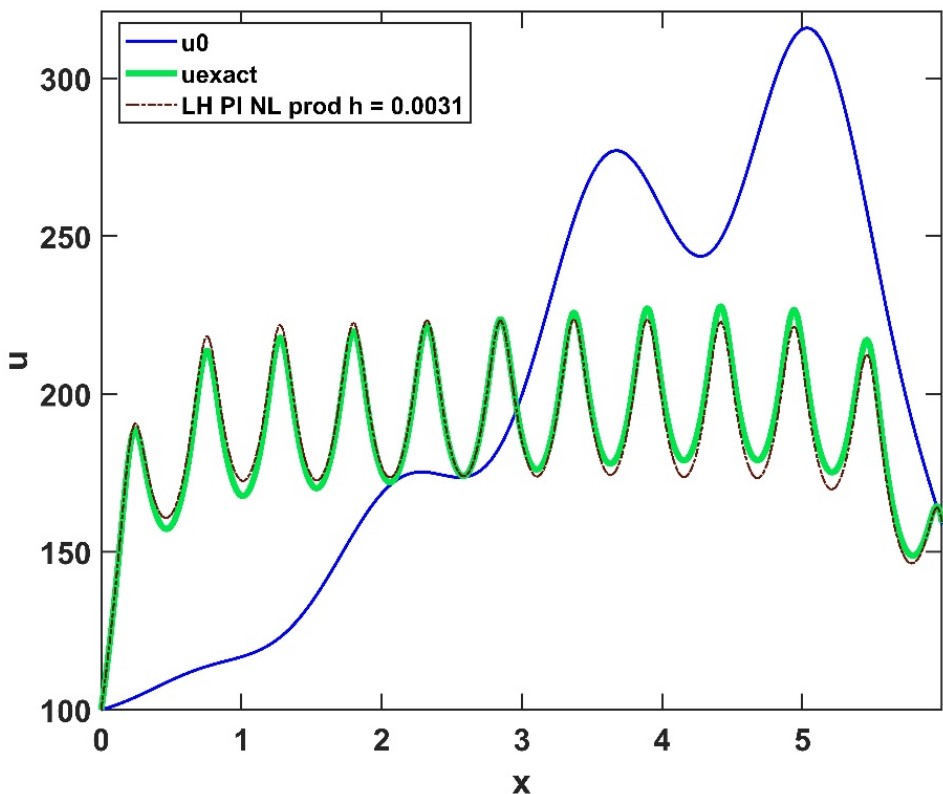

**Figure 8.** The temperature $u$ vs. the $x$ variable in the case of the initial function $u^0$, the reference solution at $t^{\text{fin}}$, and the nonlocal version of the LH method for $h = 2 \cdot 10^{-3}$.

## 6. Simulation of a Realistic Wall

### 6.1. The Structure and the Materials of the Wall

In this part, a wall segment is simulated with dimensions 1 m in the $x$ and $z$ direction and 0.2 m in the $y$ direction. As one can see in Figure 9, the following two geometries are considered:

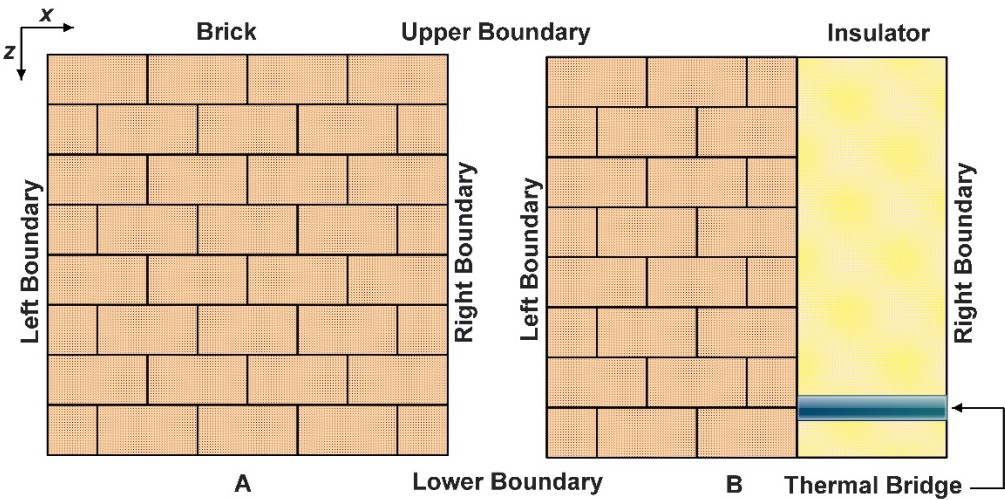

**Figure 9.** (**A**) surface of the wall, (**B**) wall with insulator and thermal bridge.

(A)   A wall's surface is examined, which is entirely built of brick.

(B)   Cross section of a wall with two layers composed of brick and rigid polyurethane foam insulator with a steel beam thermal bridge.

Real material attributes are taken into consideration, which are presented in Table 2. One should keep in mind that, although these coefficients are constants (that is, they do not change with time, space, or temperature) within a material, they have a sharp discontinuity at the boundaries of different materials.

**Table 2.** Properties of the applied materials [2].

| | $\rho$ (kg·m$^{-3}$) | $c$(J·kg$^{-1}$·K$^{-1}$) | $k$(W·m$^{-1}$·K$^{-1}$) |
|---|---|---|---|
| Brick | 1900 | 840 | 0.73 |
| Rigid Polyurethane Foam | 320 | 1400 | 0.023 |
| Steel Beam | 7800 | 840 | 16.2 |

*6.2. Mesh Construction*

In the used approximation, no heat flows and no physical quantities change in the $y$ direction, which is perpendicular to the surface of Figure 9. We use $\Delta y_i = 0.2$ m in order to obtain a realistic problem. The other two coordinates are in the unit interval, $(x, z) \in [0, 1] \times [0, 1]$, thus the surface area of the meshes is 1 m$^2$. Two kinds of meshes were constructed: the equidistant mesh has square-shaped cells, and the non-equidistant mesh has rectangular cells. We set $N_x = 100$ and $N_z = 100$ for the number of cells in the $x$ and in the z direction; therefore, $N = N_x N_z = 10{,}000$ is the total number of cells. The non-equidistant mesh contains high cells on the upper side and low ones on the lower side of the wall, as well as the cells being wide on the left side of the wall and narrow on the right. The width and height decrease gradually both in the $x$ and $z$ directions, consecutively, following a geometric series. The sum of the first $n + 1$ terms of a geometric sequence, up to the term $r^n$, $(r \neq 1)$, is

$$\sum_{k=0}^{n} ar^k = a\left(\frac{1 - r^{n+1}}{1 - r}\right) = a + ar + ar^2 + ar^3 + \ldots\ldots + ar^n. \tag{37}$$

where $n = N_x - 1 = N_z - 1$. The values $r = 0.98$ and $a = 0.0234$ are used, which give $\Delta x_1 = 0.0234$ and $\Delta z_1 = 0.0234$ on the left and the upper sides, respectively, while $\Delta z_{Nz} = \Delta x_{Nx} = 0.98^{99} \cdot \Delta x_1 = 0.00317$ on the right and lower sides. The obtained meshes are shown in Figure 10.

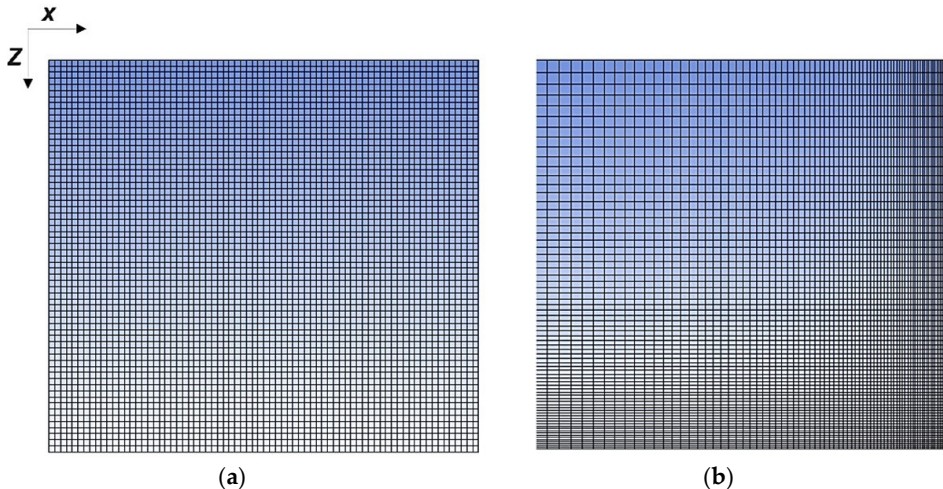

**Figure 10.** (**a**) Equidistant mesh. (**b**) Gradual change in the $x$ and $z$ directions.

In the case of the wall's surface, we apply only the equidistant grid. However, in the case of the cross section of the wall with an insulator, we apply both kinds of mesh: the equidistant and the non-equidistant. For programming simplicity, in a cross section, bricks always make up the left half of the cells, whereas the insulator (containing the thermal

bridge) makes up the right half. This means that, in the equidistant case, the thermal bridge has the same thickness and volume as the insulator, but the thickness of the insulator is smaller (0.269 m) in the non-equidistant case, as shown in Figure 9B. The horizontal position of the thermal bridge is between $x = 0.5$ m and $x = 1$ m for the equidistant mesh and between $x = 0.735$ m and $x = 1$ m for the non-equidistant one. The height of the thermal bridge is two cells (2 cm) in the $z$ direction, i.e., 0.02 m, while it is vertically positioned in rows number 20 and 21 from $z = 0.37$ m to $z = 0.39$ m.

As in our previous work, we apply a resistance–capacitance-type model [38] of heat conduction, where it is necessary to switch to cell variables. The cell temperature is defined as the temperature in the middle of the cell. We calculate the heat capacity of the cells using the formula $C_i = c_i \rho_i \Delta x_i \Delta z_i \Delta y$, while for the approximate formula for the thermal resistance in the $x$-direction, we use $R_{i,i+1} \approx \frac{\Delta x_i}{k_{i,i+1} A x}$. Here, $A x = \Delta y \Delta z_i$ is the area of the cell surface orthogonal to $x$. Therefore, the horizontal and vertical resistances in the surface simulation case (with homogenous material and uniform mesh) can be expressed as

$$R_{i,i+1} \approx \frac{\Delta x}{k_i \Delta z \Delta y} \text{ and } R_{i,i+N_x} \approx \frac{\Delta z}{k_i \Delta x \Delta y_i},$$

respectively, where the cell with the second subscript $i + N_x$ is just below the cell $i$. If the sizes and/or the material properties of the two adjacent cells are different, the horizontal and vertical resistance between cells $i$ and $i + 1$ are

$$R_{i,i+1} \approx \frac{\Delta x_i}{2k_i \Delta z_i \Delta y} + \frac{\Delta x_{i+1}}{2k_{i+1} \Delta z_i \Delta y}$$

and

$$R_{i,i+N_x} \approx \frac{\Delta x_i}{2k_i \Delta z_i \Delta y} + \frac{\Delta x_{i,i+N_x}}{2k_{i,i+N_x} \Delta z_i \Delta y}.$$

Using these quantities and approximations, one can obtain the ODE system for the time derivative of the cell variables for a general grid, as follows:

$$\frac{du_i}{dt} = \sum_{j \neq i} \frac{u_j - u_i}{R_{i,j} C_i} + q_i - K \cdot u_i - \sigma \cdot u_i^4$$

This equation is the spatially discretized form of Equation (3). The examined methods have to be adapted to this form, but actually only the conduction term is different, since all other terms are local. The quantity $r = \frac{\alpha h}{\Delta x^2}$ is not a constant anymore. In the denominators, it must be replaced by $h/2 \sum_{j \neq i} \frac{1}{C_i R_{ij}}$, while the terms such as $r(u_{i-1}^n + u_{i+1}^n)$ must also be replaced by, e.g., $h \sum_{j \neq i} \frac{u_j^n}{C_i R_{ij}}$. For instance, instead of Formula (20), now we have

$$u_i^{1/2} = \frac{u_i^0 + \frac{h}{2} \sum_{j \neq i} \frac{u_j^n}{C_i R_{ij}} + h/2 \cdot q_i - Khu_i^0/2}{1 + \frac{h}{2} \sum_{j \neq i} \frac{1}{C_i R_{ij}} - Kh/4 + \sigma h (u_i^0)^3 / 2}. \tag{38}$$

The DF and the original OEH methods can be modified similarly, but as we mentioned, the ADE method cannot, so we omit that from this section. One can read more about this formalism in our previous publications [27,37].

Since now most cells have four neighbors instead of two, as in the 1D case, the spatially nonlocal treatments cannot be applied exactly the same way. We were not able to adapt the product treatment (point 6 in Section 2.3) into this case in any simple way, so we decided to

discard it. The nonlocal average version can be adapted by simply calculating the average of the existing neighbors. For example, we can write

$$\frac{u_1^n + u_3^n + u_{N_x+2}^n}{3}(u_i^n)^2 \tag{39}$$

in the case of the second cell, which has three neighbors.

### 6.3. The Initial and the Boundary Conditions

In this part, the final time (the end of the analyzed time span) is $t_{\text{fin}} = 10{,}000$ s. The time step size is measured in seconds as well. In all cases and all boundaries, zero Neumann boundary conditions are used, which prohibits the flow of conductive heat at the boundaries:

$$\frac{\partial u}{\partial x}(x, z = 0, t) = \frac{\partial u}{\partial x}(x, z = 1, t) = \frac{\partial u}{\partial z}(x, z = 0, t) = \frac{\partial u}{\partial z}(x, z = 1, t) = 0$$

This is accomplished by setting the necessary resistances to infinity and setting the value for the matrix elements representing heat conduction through the boundary as zero.

(A) In the case of the surface area simulation, the heat transfer by radiation and convection is happening in the $y$ direction, i.e., orthogonal to the plane of Figure 9.

A linear function of the $x$ variable is applied as the initial condition:

$$u(x, z, \ t = 0) = 300 - 280x$$

For the convection heat transfer coefficient $h_c$, we have taken values from the literature [2], as one can see in Table 3. Regarding radiation, the Stefan–Boltzmann constant is a universal number $5.67 \cdot 10^{-8} \frac{W}{m^2 \cdot K^4}$. The surface is not an ideal black body; thus, we multiplied the Stefan–Boltzmann constant by the appropriate emissivity constant to obtain realistic values for $\sigma^*$. We estimate the value of $q^*$ for the heat source term, which includes the solar radiation as it is shown below. The temperature of the ambient air is taken to be $27\,^\circ\text{C} \approx 300$ K.

**Table 3.** The parameters of convection, radiation, and heat source in case of wall surface area [39].

| | $h_c\ (\frac{W}{m^2 \cdot K})$ | $\sigma^*\ (\frac{W}{m^2 \cdot K^4} \times 10^{-8})$ | $q^*_{shadow}(\frac{W}{m^2})$ | $q^*_{sunny}(\frac{W}{m^2})$ |
|---|---|---|---|---|
| All elements | 4 | 4 | 300 | 800 |

Due to the nonzero temperature $u_a$ of the air (in Kelvin), the expression $q$ also contains the convective heat gain. We can obtain the value of $q$ as follows

$$q = \frac{q^*}{c\rho\Delta y} + \frac{h_c}{c\rho\Delta y} \cdot u_{\text{a}}$$

The convective and radiative energy transfer occurs in the $y$ direction, i.e., perpendicular to the surface. Consequently, these terms are proportional to the element's free surface area, which is $\Delta x \Delta z$ here. This information yields the following values for the coefficients in Equations (2) and (3):

$$K = \frac{h_c}{c\rho\Delta y}, \quad \sigma = \frac{\sigma^*}{c\rho\Delta y}.$$

We assumed that the lower-half side of the surface is in the shade, resulting in much less incoming heat there. Specifically, we have

-　the first portion of N (sunny side):

$$q = \frac{1}{c\rho\Delta y} \times 800\frac{W}{m^2} + \frac{h_c}{c\rho\Delta y} \times 300\text{K};$$

- the second portion of $N$ (shaded side):

$$q = \frac{1}{c\rho\Delta y} \times 300\frac{W}{m^2} + \frac{h_c}{c\rho\Delta y} \times 300\text{K}.$$

(B) In the case of the simulation of the cross-sectional area with the thermal bridge, the interior components cannot absorb or lose heat via convection, radiation, and the heat source. According to Table 4, elements on the right and left sides may transfer heat in the $x$ direction through radiation and convection.

**Table 4.** The convection, radiation, and heat source characteristics on both sides of the wall components in the case of the cross-sectional area.

|  | $h_c(\frac{W}{m^2 \cdot K})$ | $\sigma^* (\frac{W}{m^2 \cdot K^4} \times 10^{-8})$ | $q^* (W)$ |
|---|---|---|---|
| Right Elements | 2 | 5 | 500 |
| Left Elements | 4 | 4 | 500 |

The coefficient values in our equations are obtained as follows:

$$K = \frac{h_c}{c\rho\Delta x}, \sigma = \frac{\sigma^*}{c\rho\Delta x}, q = \frac{q^*}{c\rho\Delta x} + \frac{h_c}{c\rho\cdot\Delta x}\cdot u_a$$

The ambient air temperature is taken to be 20 °C ≈ 293 K and 40 °C ≈ 313 K on the brick and the insulation side (inside and outside of the building), respectively. It gives the following convection and radiation heat sources for left and right elements:
In terms of the left-hand side:

$$q = \frac{1}{c\rho} \times 500\frac{W}{m^2} + \frac{h_c}{c\rho \cdot \Delta x} \times 293\text{K}$$

In terms ofthe right-hand side:

$$q = \frac{1}{c\rho} \times 500\frac{W}{m^2} + \frac{h_c}{c\rho \cdot \Delta x} \times 313\text{K}$$

In this case, a linear function of the $z$ variable is used for the initial condition:

$$u(x, z,\ t = 0) = 313 - 293z.$$

Until this point, all temperatures were close to room temperature. However, for significantly larger temperatures, the nonlinear radiation term has a much stronger effect. Thus, in the following point, we try to simulate a case, e.g., a furnace, where the temperature on the left side of the wall is much higher than on the right side.

(C) In case of the cross section of the wall with high temperatures, the geometry is similar to the previous case. The concrete values of the constants change according to Table 5.

**Table 5.** The heat source, convection, and radiation characteristics are on both sides of the wall components in the case of a cross section of the wall.

|  | $h_c$ ($\frac{W}{m^2 \cdot K}$) | $\sigma^*$ ($\frac{W}{m^2 \cdot K^4} \times 10^{-8}$) | $q^*$ (W) |
|---|---|---|---|
| Right Elements | 2 | 5 | 500 |
| Left Elements | 25 | 4 | 3500 |

The "ambient" air temperature inside the furnace is taken to be 227 °C $\approx$ 500 K. This yields the following convection and radiation heat sources for right and left elements, respectively:

$$q = \frac{1}{c\rho} \times 500 \frac{W}{m^2} + \frac{h_c}{c\rho \cdot \Delta x} \times 303 \text{K}$$

and

$$q = \frac{1}{c\rho} \times 3500 \frac{W}{m^2} + \frac{h_c}{c\rho \cdot \Delta x} \times 500 \text{K}.$$

In this case, a linear function of the $x$ variable is used again for the initial condition

$$u(x, z, t = 0) = 500 - 303x.$$

In this section, we used Equation (27) to calculate the maximum error where the ode15s solver was employed to obtain the reference solution. We calculated the maximum time step size (CFL limit, above which the explicit Euler time integration becomes unstable) and the stiffness ratio in the usual way [28,37], considering only the conduction term. Table 6 shows the values of these quantities for different cases.

**Table 6.** The CFL limit and the stiffness ratio quantities for the different cases.

|  |  | CFL Limit | Stiffness Ratio |
|---|---|---|---|
| Surface |  | 55.78 | $8.1 \times 10^3$ |
| cross section with thermal bridge | equidistant | 5.01 | $5.167 \times 10^5$ |
|  | non-equidistant | 2.28 | $4.29 \times 10^5$ |

*6.4. Results for the Surface of the Wall*

Experiment 5. We begin with the examination of the effect of convection only, so no radiation presents. A one-layer brick wall is simulated here (see Figure 10A), and only the equidistant mesh is used. The initial and boundary conditions as stated in point (A) above are considered, with the exception that $\sigma^*$ and $q^*$ are set to zero, so there is no sunshine in this experiment. The maximum errors as a function of time step size are shown in Figure 11 for all methods of Section 2.2. For smaller time step sizes, the mixed treatment of the convection terms with 50% weight for the PI and the 'inside' treatment is the most accurate. It clearly has a second-order convergence, in accordance with the results of the previous sections. Thus, from this point, the convection term is always treated in this optimal way. For some large time step sizes, however, the original hopscotch method is slightly more accurate, which may be due to the low stiffness ratio of the problem. In Figure 12, we show the initial and final temperature distributions.

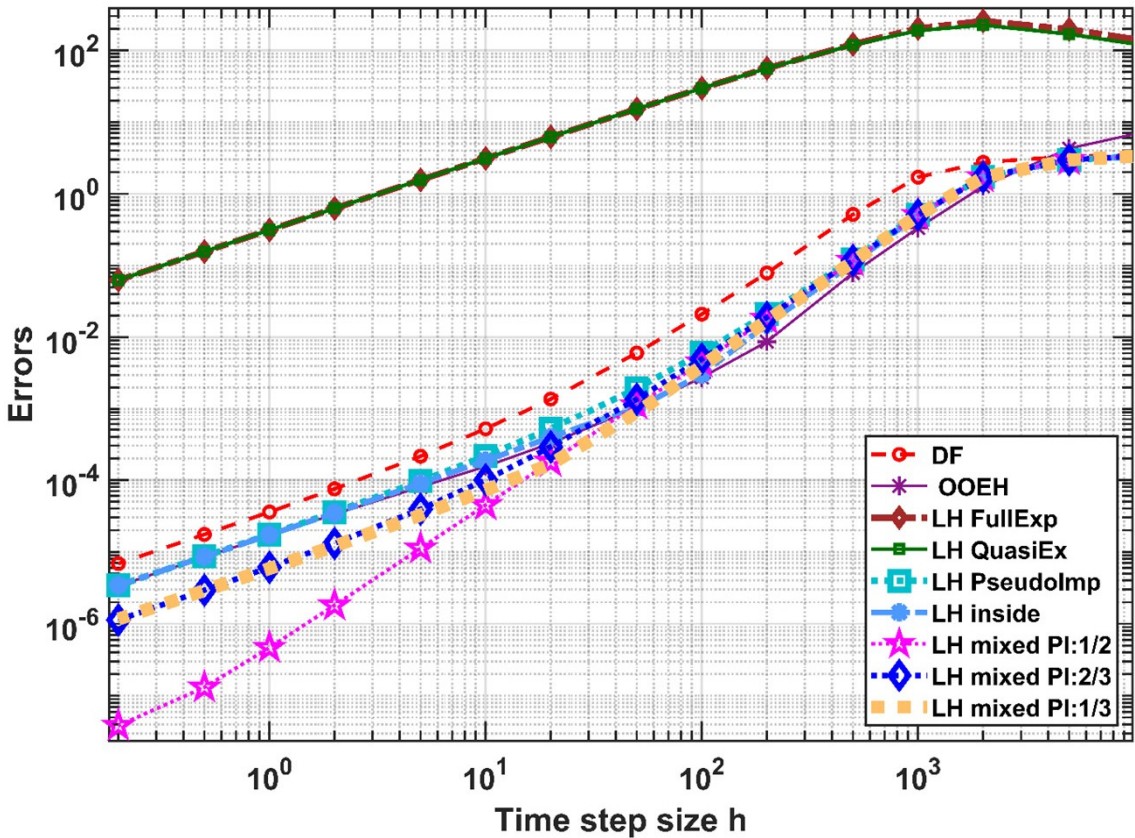

**Figure 11.** The maximum errors as a function of the time step size *h* for the nine versions of the method in the case of the surface area (Experiment 5). We emphasize that the labels referring to the treatments are now for the convection term.

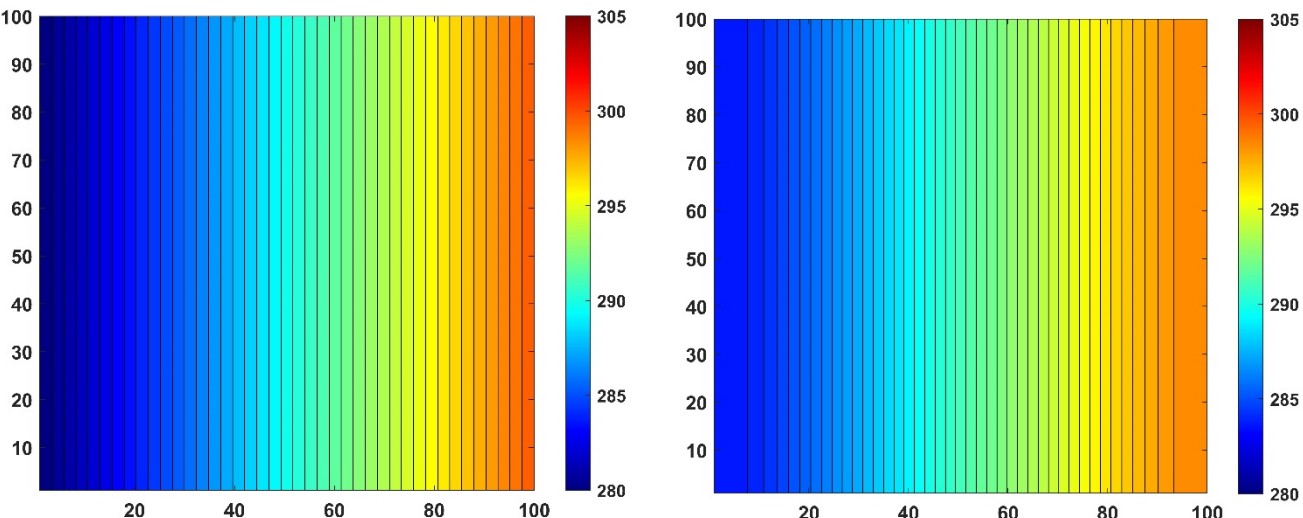

**Figure 12.** The temperature distribution contour in Kelvin units for the equidistant mesh at initial (**left**) and final time (**right**), in the case of a surface area (Experiment 5). The coordinates in cm units are represented by the numbers on the vertical and horizontal axes, which are the cell indices as well.

Experiment 6. Now, the surface of the brick wall is simulated with radiation, where conditions in point (A) and the values from Table 3 are used. The errors are shown in Figure 13, and the temperature distributions are shown in Figure 14. The influence of the initial condition, as well as the shadow, on the lower part of the wall is clearly visible.

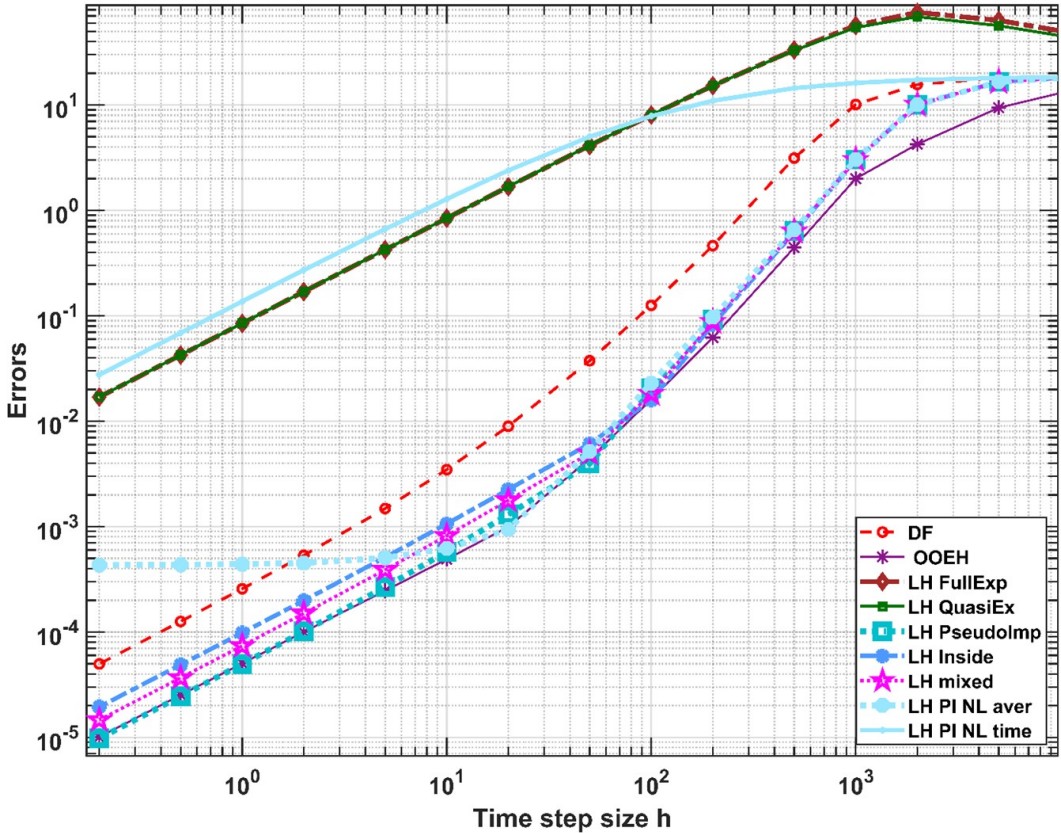

**Figure 13.** The maximum errors as a function of the time step size for the nine examined methods in the case of a surface area (Experiment 6). We emphasize that the different treatments now refer to the radiation term.

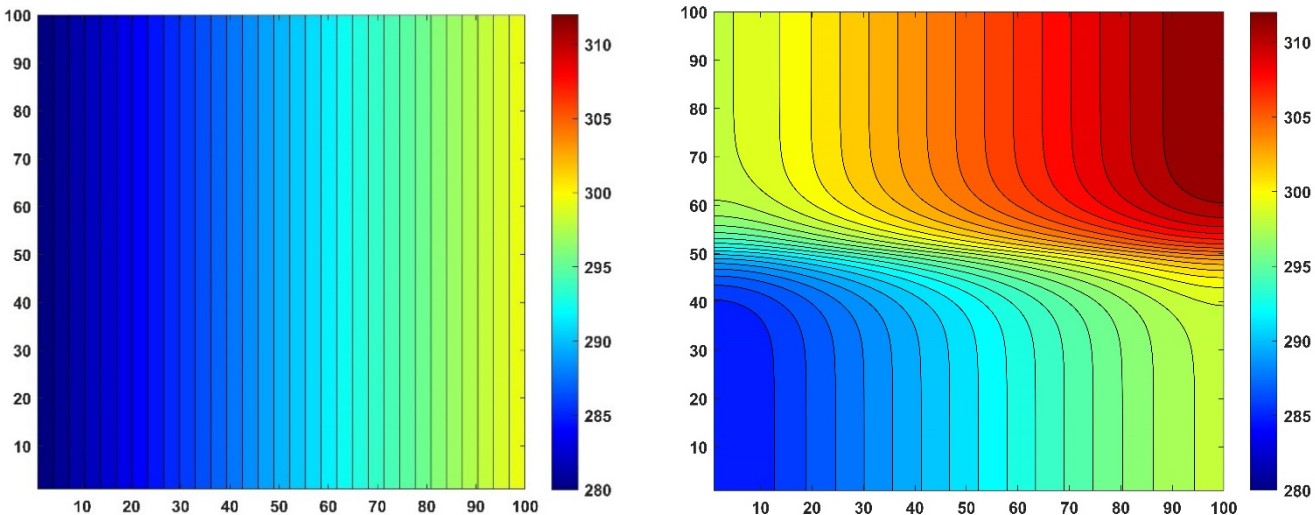

**Figure 14.** The temperature distribution contour in Kelvin in the case of Experiment 6 at initial (**left part**) and final time (**right part**), in the case of a surface area. The coordinates in cm units are represented by the numbers on the vertical and horizontal axes of the contours, which are the cell indices as well.

*6.5. Results for the Cross Section of the Insulated Wall with Thermal Bridging*

Experiment 7. In this case, the initial and boundary conditions listed in point (B) are applied to the multilayer wall with the equidistant mesh. The errors are plotted in Figure 15. The temperature distribution contour for the initial and final time moments are

presented in Figure 16. The temperature on the right side of the wall rises because of the higher outside temperature, albeit the insulator allows this heat to enter the wall at a very slow rate.

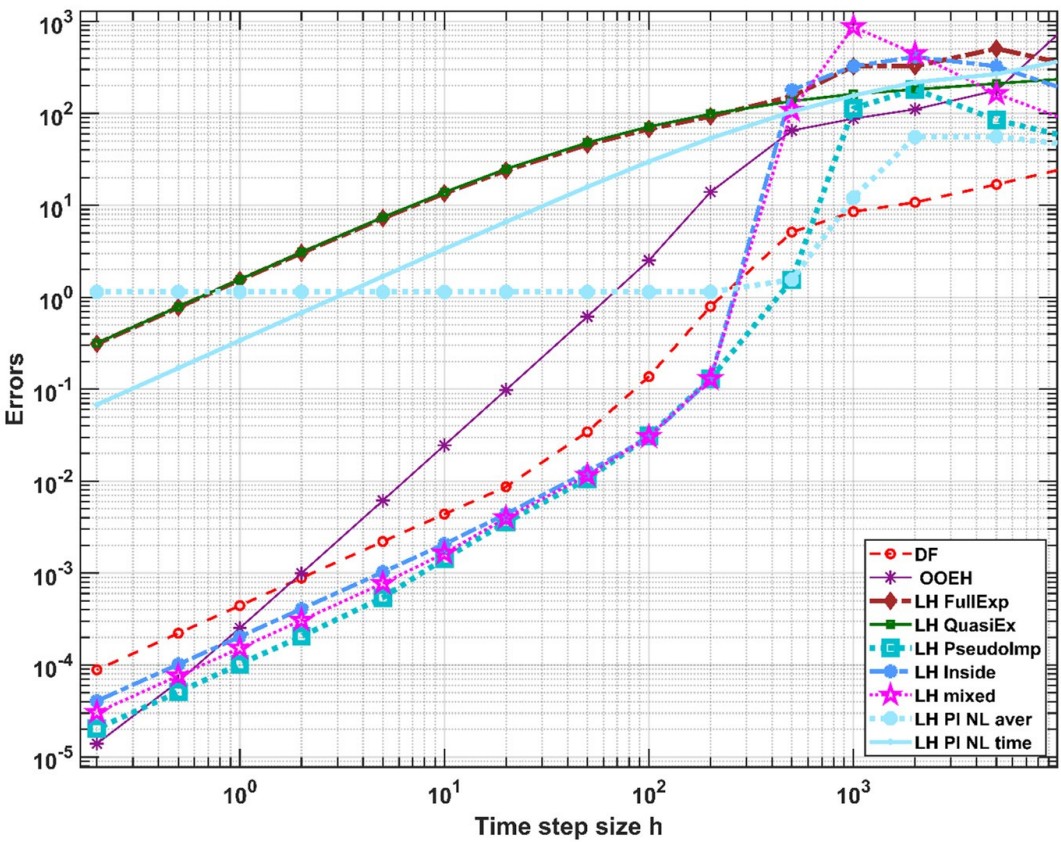

**Figure 15.** The maximum errors as a function of *h* for the equidistant mesh (Experiment 7) in the case of convection and radiation boundary conditions.

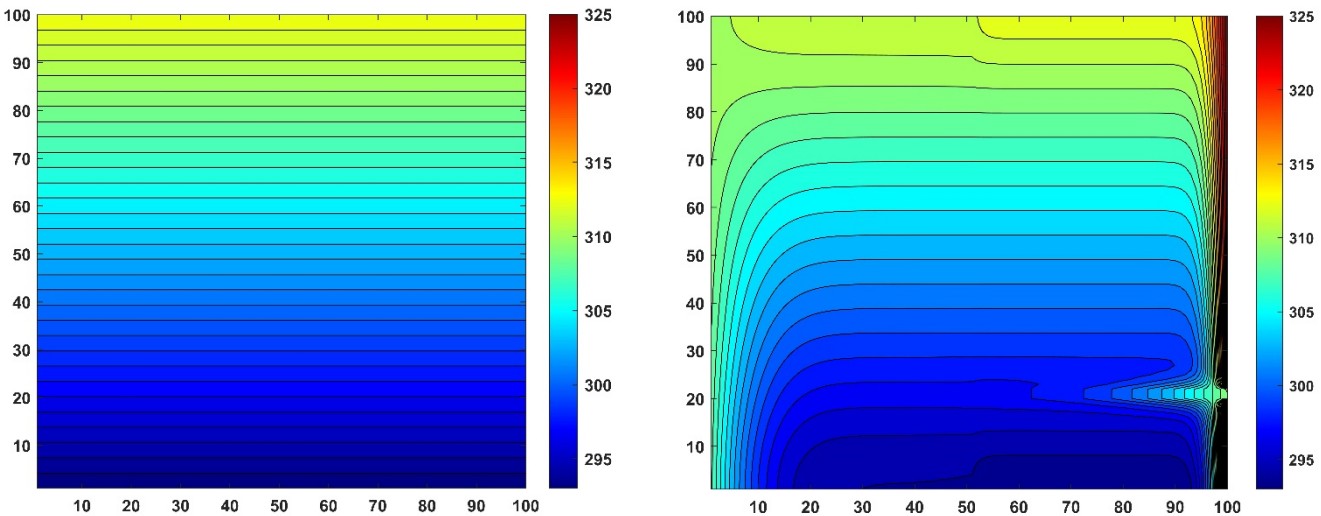

**Figure 16.** The temperature distribution contour in Kelvin units at the initial (**left**) and the final time (**right**) in the case of Experiment 7 (equidistant mesh).

Experiment 8—non-equidistant mesh. Everything is the same as in Experiment 7, but the mesh is replaced by the non-equidistant one presented in Figure 10b. The errors, as well as the temperature contours, are shown in Figures 17 and 18, respectively.

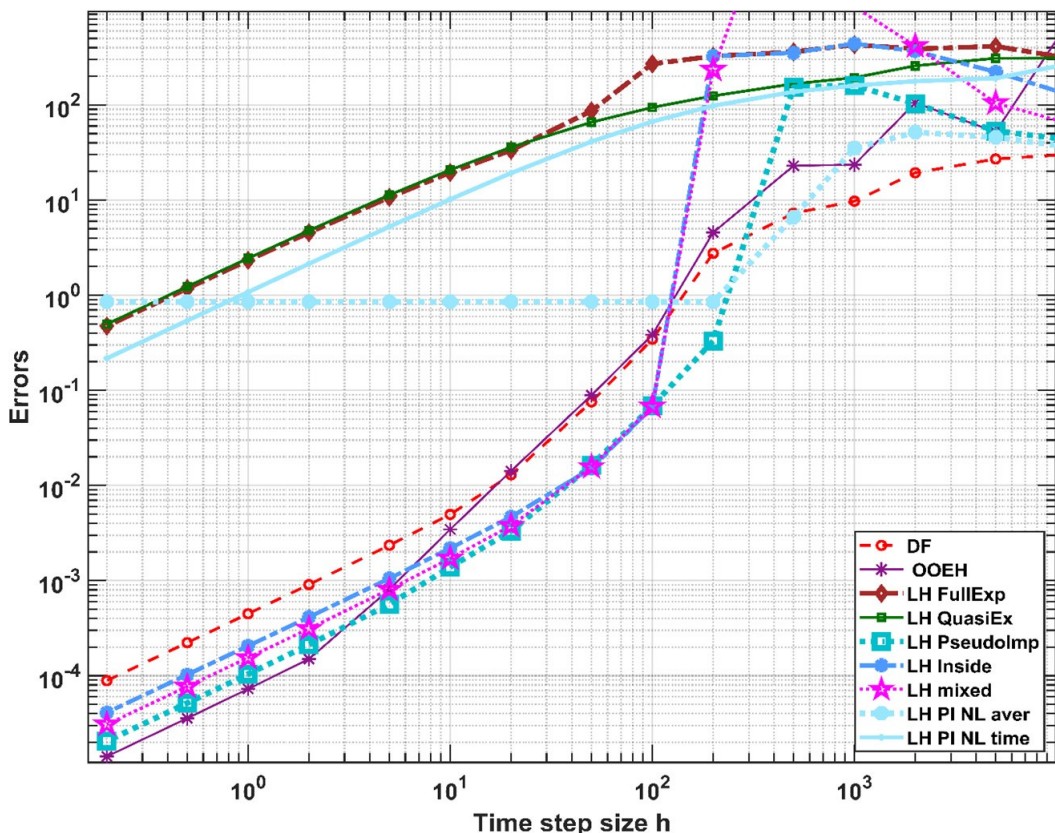

**Figure 17.** The maximum errors vs. the time step size *h* for the non-equidistant mesh (Experiment 8) in the case of convection and radiation boundary conditions.

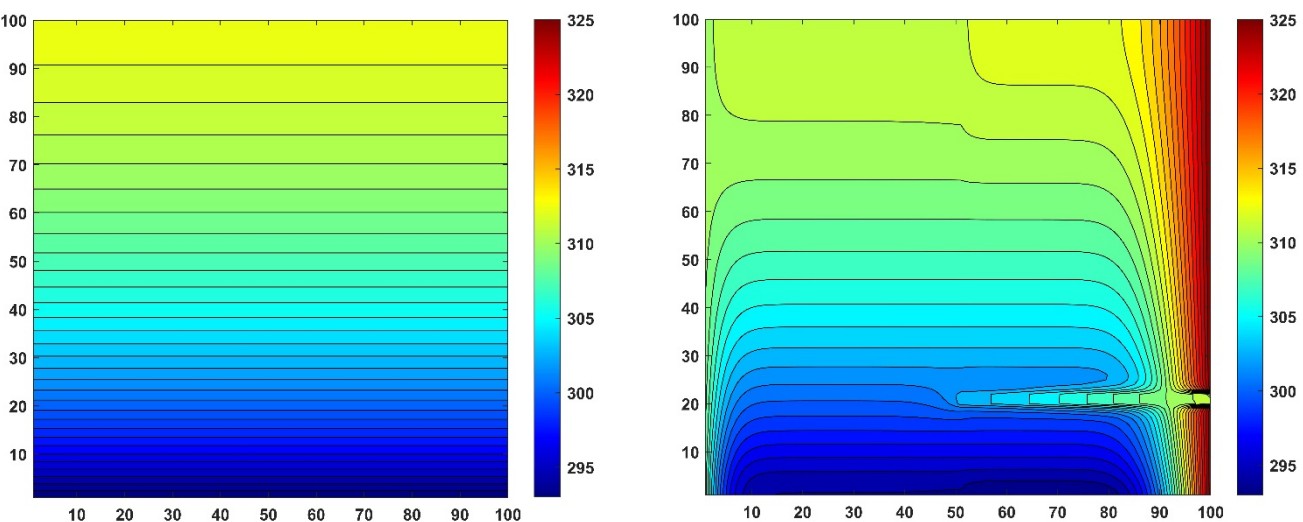

**Figure 18.** The distribution contour of the temperature in Kelvin for the non-equidistant mesh at initial (**left**) and final time (**right**), in the case of the multilayer cross-sectional area (Experiment 8). The numbers on the vertical and horizontal axes of the contours are the coordinates in cm units.

Experiment 9—high-temperature boundary conditions, non-equidistant mesh.

In this case, a multilayer wall with the thermal bridge is simulated using the non-equidistant mesh. The linear initial and the Neumann boundary conditions of point (C) are used. The errors are plotted in Figure 19. One can see that the mixed and inside versions produce rather large errors, and they are clearly outperformed by the PI treatment. The

temperature distribution contour for the initial and final time moments are presented in Figure 20.

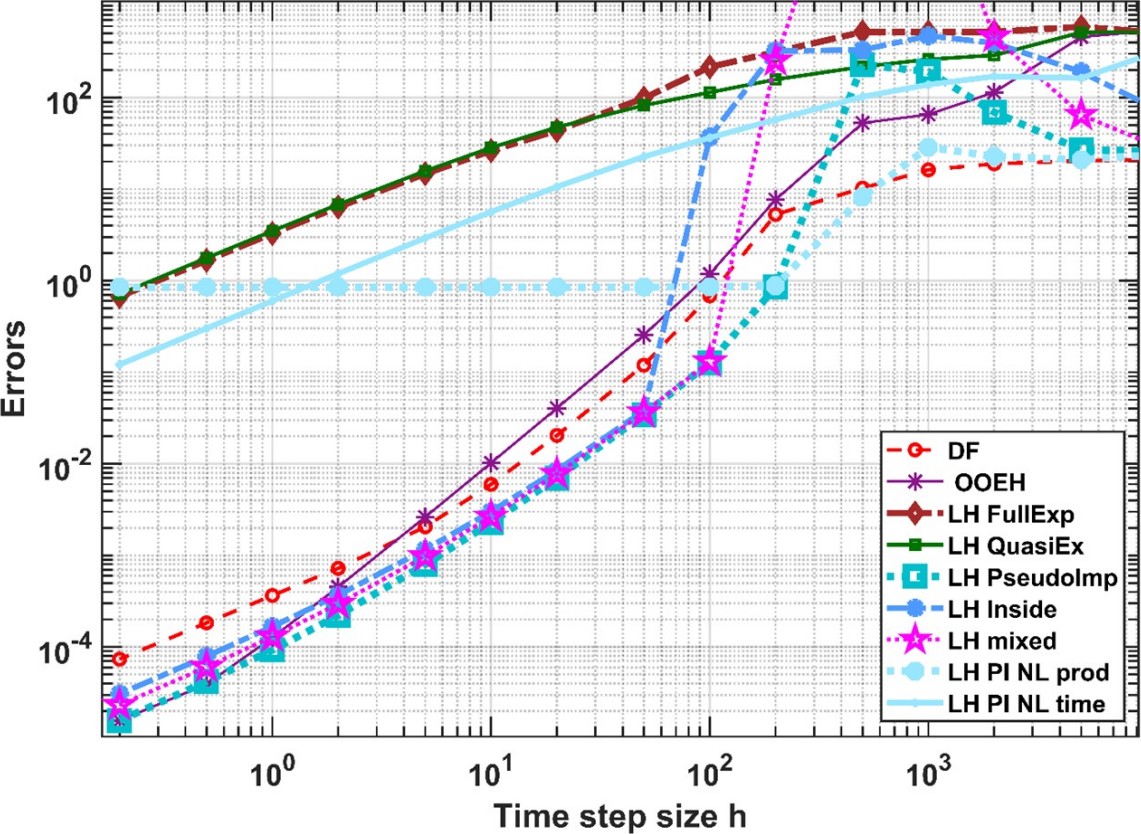

**Figure 19.** The maximum errors as a function of the *h* for Experiment 9 in the case of convection and radiation with high-temperature boundary conditions.

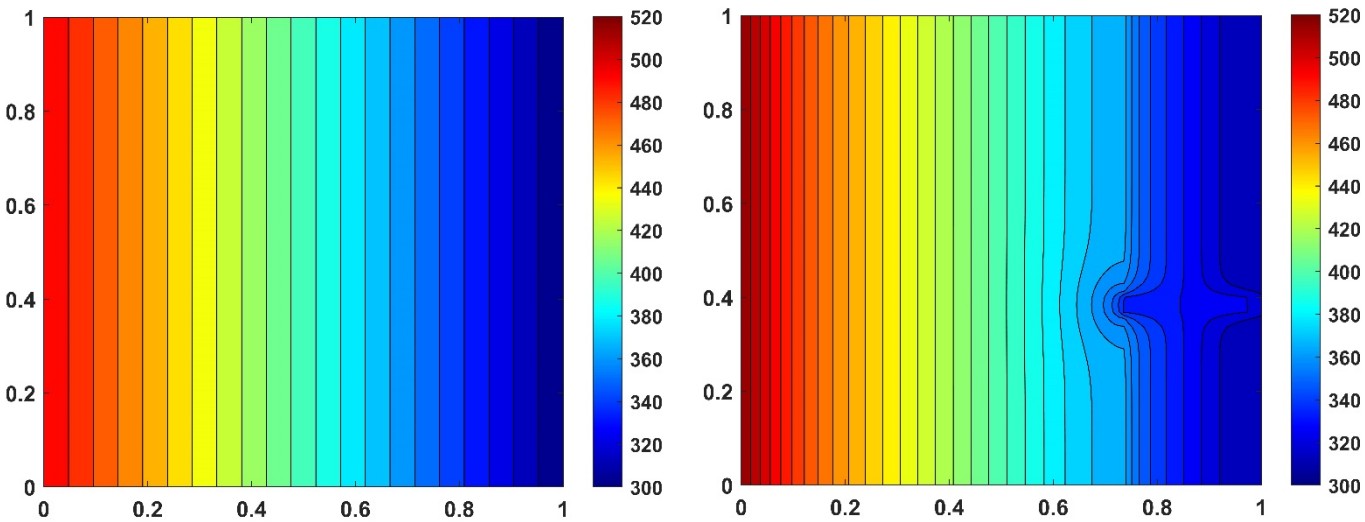

**Figure 20.** The distribution contour of the temperature in Kelvin units for the equidistant mesh at initial and final time (**left** and **right** figure, respectively), in the case of Experiment 9. The numbers on the axes are the coordinates in meter units.

### 7. Discussion and Summary

We have studied several implementations of the free convection and radiation terms using the leapfrog-hopscotch method, which had originally been optimized to solve the heat conduction equation. In our last paper [31], we proposed the fully pseudo-implicit treatment of the convection term. In this paper, it was observed that, usually, the best performance is achieved when the convection term is treated in a mixed way, i.e., taking into account 50% at the old and 50% at the new time level. The order of temporal convergence reaches two only for this optimal version, which was also proven by the calculation of the truncation errors. The unconditional stability of this version was also proven by von Neumann analysis in the linear case (conduction + convection).

On the other hand, according to the numerical experiments, the radiation term should be taken into account fully in the pseudo-implicit way. In this case, one of the four powers is taken into account at the new time level, so the term turns up only in the denominator, which ensures very good stability properties. We performed four numerical experiments in the one-dimensional case, and then another five to simulate heat transfer of a realistic wall. The proposed algorithm performs quite well, even when the CFL limit for the mainstream explicit methods is rather low.

In our next work, we are going to make extensive measurements of the running times of the best algorithms here, and compare them to those of other available methods and solvers, e.g., the built-in solvers of MATLAB and Ansys Fluent. Then, the best methods will be used for real simulations, aiming to design and optimize buildings with better thermal properties. The studied methods can also be used to simulate multiphysics problems, e.g., when the drift diffusion of the charge carriers in semiconductors is coupled with heat transfer, or when underground heat transfer is assisted by groundwater flow [39].

**Author Contributions:** Conceptualization, methodology, supervision, and resources, E.K.; analytical investigation, J.M., software, numerical investigation, and visualization, A.H.A. and I.O.; writing—original draft preparation, E.K. and A.H.A.; writing—review and editing, I.O. All authors have read and agreed to the published version of the manuscript.

**Funding:** This research received no funding.

**Data Availability Statement:** Data available on request from the authors.

**Conflicts of Interest:** The authors declare no conflict of interest.

### Nomenclature

| Symbols | | Greek Symbols | |
|---|---|---|---|
| $c$ | Specific heat (kJ/kg.K) | $\alpha$ | Thermal diffusivity ($m^2$/s) |
| $h$ | Time step size (sec) | $\Delta$ | Difference |
| $h_c$ | Heat transfer coefficient (W/$m^2$.K) | $\rho$ | Mass density (kg/$m^3$) |
| $K$ | Convection coefficient (1/sec) | $\sigma$ | Coefficient of the radiation term ($sec^{-1}$ $K^{-3}$) |
| $k$ | Thermal conductivity (W/m.K) | $\sigma^*$ | realistic values of non-black body (W/$m^2$.$K^4$) |
| $Q$ | Heat transfer rate (W) | *Subscripts* | |
| $q^*$ | heat generation (W/$m^2$) | $a$ | Ambient air |
| $q$ | Heat source rate(1/K) | sunny | Sunny surface |
| $t$ | time (sec) | shadow | Shadow surface |
| $u$ | Temperature ($K^o$) | | |

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
