# Peer review of "Testing Some Different Implementations of Heat Convection and Radiation in the Leapfrog-Hopscotch Algorithm"

_algorithms, doi:10.3390/a15110400_

Round 1

Reviewer 1 Report

The paper generally is written well. The explanation in section 2 is good. Only a sentence in Line 116 is not fully comprehensible. The meaning of the terms alpha, q_i and h should be explicitly stated (although can be guessed in the context). For the implementation of conv/rad-terms an overview table could help, where names given in the result figures (e.g. LH mixed PI:1/2) are listed with full name and eqn number.

The results section 6 should be better organized. Which initial and boundary conditions (A, B, C) are used in which experiments? The figures and fiugure captions should be reordered, so that it will become clear which experiment with which BC has which error plot and which temperature distribution. The colour code of the temperatures should not be changed between initial and final distribution.

Maybe I did not understand the simulation case A (experiment 5): Is there a heat transfer in y-direction (on both sides or one side y=0 and/or y=0.2m? Is the surface temperature in figure 12 correct? Is it in the x-z-plane? Why is there no impact of the shading (see lines 498 ff) - the lower part should be cooler, and the initial temperature distribution in x should vanish in my understanding, changing into a distribution in z. Please make the assumptions and results more transparent.

For the error - is it still true that the reference for the error also in section 6 is the MATLAB software ode15s solver? Using eqn. (27) ? In section 3 it is said that that is being used only for sections 3 and 5.

Reviewer 2 Report

The manuscript studied the leapfrog-hopscotch scheme. They discussed the convection and radiation terms in several ways to achieve the optimal implementation with the aim of solving the nonlinear heat conduction-convection-radiation equation. The authors well investigated the different terms of the equation and proposed an alternative solution. Several formulations are presented and well discussed..

Further comments:

Please correct the format of the authors according to journal’guideline.

I suggest to introduce a nomenclature table  where all parameter can be explained and their measurement units can be defined.

The Introduction has to be more detailed. Please read and introduce these manuscripts in line 45-48 considering differential equation used to solve transfer heat into the aquifer in finite difference approach:

-         Alberti, Angelotti, Antelmi, La Licata, Borehole Heat Exchangers in aquifers: simulation of the grout material impact, X Convegno dei Giovani Ricercatori di Geologia Applicata 2016, Università di Bologna, 2016, Rendiconti Online Società Geologica Italiana, Vol. 41 (2016), pp. 268-271, doi: 10.3301/ROL.2016.145

-         Angelotti, Alberti, Licata, Antelmi, Borehole Heat Exchangers: heat transfer simulation in the presence of a groundwater flow, Journal of Physics: Conference Series Vol. 501 (2014) 012033, doi:10.1088/1742-6596/501/1/012033

Lines 99-103. Please better discuss what the bipartite mesh is.

I suggest to revise English with a mother tongue teacher

Line 401. A symbol is not clear.

Figure 11, 12 and 13. It is a mix of figure, they need to be re-formatted and presented again in another way

Figure 14 is missing

Discussion needs to be more detailed. Please discuss in a clearer way which is the innovation of this work and the process you followed to reach this aim. What are the future perspectives?

Round 2

Reviewer 2 Report

The authors applied all the suggestions and improved the quality of the manuscript.